# Exploring Old Data with New Tricks: Long-Term Monitoring Indicates Spatial and Temporal Changes in Populations of Sympatric Prairie Grouse in the Nebraska Sandhills

**Danielle J. Berger** [1,*,†], **Jeffrey J. Lusk** [2], **Larkin A. Powell** [1] and **John P. Carroll** [1]

[1] School of Natural Resources, University of Nebraska-Lincoln, Lincoln, NE 68583, USA
[2] Nebraska Game and Parks Commission, Lincoln, NE 68503, USA
* Correspondence: danielle.j.berger@gmail.com
† Current address: Quinney College of Natural Resources, Utah State University, Logan, UT 84322, USA.

**Abstract:** The contiguous grasslands of the Sandhills region in Nebraska, USA, provide habitat for two sympatric, grassland-obligate species of grouse, the greater prairie-chicken (*Tympanuchus cupido pinnatus*) and the plains sharp-tailed grouse (*Tympanuchus phasianellus jamesi*). Collectively referred to as prairie grouse, these birds are monitored and managed jointly by wildlife practitioners who face the novel challenge of conserving historically allopatric species in shared range. We reconstructed region-wide and route-specific prairie grouse population trends in the Sandhills, using a 63-year timeseries of breeding ground counts aggregated from old reports and paper archives. Our objective was to repurpose historical data collected for harvest management to address questions pertinent to the conservation of prairie grouse, species whose populations have declined precipitously throughout their respective ranges. Because we cannot change the sampling protocol of historical data to answer new questions, we applied 3 different methods of data analysis—traditional regional mean counts used to adjust harvest regulations, spatially implicit, site-specific counts, and spatially explicit trends. Prairie-chicken populations have increased since the 1950s, whereas sharp-tailed grouse populations have remained stable or slightly declined. However, each species exhibited unique shifts in abundance and distribution over time, and regional indices masked important aspects of population change. Our findings indicate that legacy data have the capacity to tell new stories apart from the questions they were collected to answer. By integrating concepts from landscape ecology—a discipline that emerged decades after the collection of our count data began—we demonstrate the potential of historical data to address questions of modern-day conservation concern, using prairie grouse as a case study.

**Keywords:** abundance; demography; distribution; game bird; Great Plains; space use; *Tympanuchus cupido*; *Tympanuchus phasianellus*

## 1. Introduction

Conservation efforts are most often implemented reactively [1], supported by data collected from struggling populations that may no longer fulfill their ecological roles and are subject to the negative effects of low numbers [2,3]. Recently, wildlife practitioners have come to appreciate the value of proactive conservation, which emphasizes protecting populations before they decline rather than focusing on species recovery [4]. While it may seem simple to conserve species in locations where their populations are doing well, managers are confronted with a data gap. Most conservation-motivated research efforts are concentrated in locations where populations are imperiled and focus on drivers of population decline [5]. However, understanding why some populations are struggling does not necessarily provide managers with insight into factors that promote population stability and growth. To promote effective conservation, we also need to study well-performing populations to understand why they are doing well.

The greater prairie-chicken (*Tympanuchus cupido pinnatus*; hereafter, prairie-chickens) and plains sharp-tailed grouse (*Tympanuchus phasianellus jamesi*; hereafter, sharptails)—collectively known as prairie grouse—provide an ideal example of the well-performing population data gap in conservation. Both species have experienced precipitous population declines across their respective ranges and are subspecies of conservation concern in the Great Plains [6,7]. Prairie grouse research efforts have primarily focused on regions where these species are declining to uncover the drivers of downward population trends [8–11]. Studies of declining populations have provided important insights into how grassland habitat loss and degradation resulting from agricultural intensification [11,12], grazing practices [13], anthropogenic development [14], and woody encroachment [15] have negatively affected prairie grouse demographic rates and abundance, providing managers with vital information to triage struggling populations. In contrast, few studies have focused on stable prairie grouse populations in intact grasslands, such as those found in the Sandhills of Nebraska [13,16–18], which may provide different, but equally vital information for local and range-wide species conservation.

Large, stable populations of prairie grouse are likely subject to different environmental, anthropogenic, and social drivers than their declining, but more intensively studied counterparts. The Sandhills, a 50,000 km$^2$ mixed-grass prairie in northcentral Nebraska, is the largest remaining contiguous grassland in North America [19], minimizing the consequences of habitat loss and environmental degradation. Nebraska is also one of the few states where harvest of both prairie grouse species is permitted [20]. Historically, prairie-chickens and sharp-tailed grouse were allopatrically distributed but have recently come to occupy a shared range in the Sandhills of Nebraska, resulting in a unique social environment [21]. Prairie-chickens and sharp-tailed grouse may compete for common resources or have differing resource needs that must now be met in common habitat [17]. Because of their abundance in the Sandhills [6,22], prairie grouse are also less likely to be subject to the problems that plague small populations, including demographic stochasticity and Allee effects [2,23]. As economically, ecologically, and culturally important species in the Great Plains, prairie grouse are frequently targeted for conservation action [24]. In the face of increasing environmental variability, wildlife managers are coming to recognize the importance of regional conservation planning and maintaining large, stable core populations for species persistence [25]. However, large-scale, data-driven conservation planning requires knowledge of species trends and their drivers for all populations, including historically stable populations where there is often a data deficit.

The challenge for budget-constrained managers is to find a cost-effective strategy to assess stable and declining populations. The solution for prairie grouse, which may also apply to other game species, is not collecting new data but repurposing long-term monitoring data to answer conservation-focused questions. The Nebraska Game and Parks Commission (NGPC) first began collecting species-specific prairie grouse spring breeding ground count (hereafter, SBGC) data in the 1950s to monitor population trends and inform adjustments to prairie grouse bag limits, season length, and boundaries of huntable zones [22]. While the intent of the SBGC data collection was to inform harvest regulations, the full 60+ year time series serves as a longitudinal population study with broad spatial coverage, matching the spatiotemporal scale of greatest concern for conservation planning. Unlike the quasi- or natural experimental designs of short-term, intensive prairie grouse research, SBGC data were collected to monitor population trends rather than gain understanding of the ecological processes that give rise to the observed patterns. However, the length of the SBGC timeseries and broad spatial coverage of survey transects likely capture sufficient environmental variation to provide valuable insight into the processes that shape prairie grouse population trends. The large spatiotemporal scale of the SBGC data may also capture important population processes for prairie grouse that are missed in short-term, small-scale studies [26].

Long-term harvest monitoring data have value for addressing questions of conservation concern, but practitioners need to be mindful of how the original intent of the

data collection and its corresponding survey design may influence alternate usage [27–29]. Breeding ground survey transects were placed opportunistically along 20-mile (~32 km) stretches of rural roads throughout the Sandhills with the number and spatial distribution of survey transects varying with time and staffing. Historically, counts from individual transects were aggregated as spatial replicates to provide an index of mean abundance for a species in a given year [30]. The SBGC survey protocol reflects NGPC's need for a general prairie grouse population trend to inform hunting regulations that would be applied uniformly across a large section of the state. However, if NGPC's goal is to use SBGC data to uncover long-term drivers of prairie grouse population trends, we should consider the accuracy of the mean trend and how the spatial context of the survey routes may have influenced observed counts.

Breeding ground monitoring protocols were developed prior to the recognition of the Modifiable Areal Unit Problem, which describes sampling issues in a heterogenous environment [28]. The response variable—the number of prairie chickens observed along a 20-mile transect—is measured at locations with different underlying habitat attributes and is likely influenced by some of those environmental characteristics. If the position of the sampling unit boundaries was shifted in geographic space (as was the case for NGPC's surveys when the spatial configuration of transects changed between years), the number of prairie grouse counted would likely be different than if the survey was conducted at the original sampling location because of differences in the underlying environmental space. Transects are therefore "modifiable" sampling units [27]. While the differences among sampling units are what will eventually help to disentangle the processes driving population trends, the Modifiable Areal Unit Problem complicates computing the aggregate regional trends likely to spur conservation action. If survey routes in heterogeneous habitat are not substitutable, changes in the number and spatial configuration of transects between years could produce population trends that reflect artifacts of sampling design rather than true fluctuations in prairie grouse numbers. Repurposing harvest data requires careful reconsideration of how we model abundance trends to ensure they reflect biological processes rather than our methods of sampling.

Our goal was to create the first complete time series of Nebraska's SBGC data from 1956 to 2018 and repurpose a harvest-oriented data source to explore trends that will inform critical prairie grouse conservation questions. Our first objective was to present longitudinal, species-specific SBGC trends for prairie grouse in the Sandhills to contribute to state and regional conservation efforts [31,32]. Our second objective was to compare population growth rates of prairie-chickens and sharptails since 1956. As historically allopatric species, differences in growth rates in shared range suggest that interspecific competition, species-specific population drivers, or other biological processes may constrain management of the two bird species in common habitat. Our third objective was to evaluate how different methods of data aggregation influence our understanding of prairie grouse population trends in the Sandhills, comparing pooled, transect-specific (spatially implicit), and spatially explicit approaches. We considered the merits of each approach for addressing sampling issues such as the Modifiable Aerial Unit Problem and providing population trends at spatiotemporal scales most relevant for conservation decision-making.

## 2. Materials and Methods

### 2.1. Study Area

The Sandhills are a 50,000 km$^2$ grassland in north-central Nebraska characterized by vegetation-stabilized sand dunes interspersed with subirrigated meadows and wetlands [33]. Low annual precipitation and high evapotranspiration rates coupled with sandy entisol soils characterize the Sandhills as a semi-arid region [34]. The mean annual precipitation in the Sandhills is 580 mm in the east and 430 mm in the west [35]. The east to west precipitation gradient drives a progression of vegetation communities in the Sandhills, from dominant tallgrass prairie species in the east to mid- and shortgrass species that prevail in the west [36].

The topography of the Sandhills follows an east to west gradient, with western dunes having the greatest relief (120 m high) [33]. Elevation, capturing distance to the water table in the Sandhills, also shapes the distribution of vegetation communities. Upland areas are comprised of warm-season tallgrasses, interspersed with mid- and short-grasses [37]. In subirrigated meadows and wetlands—flatter areas between the dunes where the groundwater is at or near the soil surface—the vegetation is a mixture of cool and warm-season grasses, grass-like plants, and some woody species [37].

The vegetation structure and composition of upland areas was historically shaped by fire and roaming herds of grazing bison (*Bison bison*) [33]. Today, the grazing management practices of beef cattle producers are the strongest determinant of vegetation structure and community composition [33]. The sandy, unstable soils, dry climate, and undulating terrain of the Sandhills rendered most cultivation impractical, and only about 5% of the total land area has been placed in crop production [38]. Afforestation via woody encroachment, often from planted eastern redcedar (*Juniperus virginiana*) windbreaks, has resulted in a loss of grassland area. Woody encroachment likely poses the greatest future threat to the integrity of the Sandhill's intact upland prairie system [39].

### 2.2. Breeding Ground Counts

Since 1956, the NGPC has monitored prairie grouse abundance using annual count surveys on the spring breeding grounds (Figure 1). Between April 1 and 9 on clear mornings with little to no wind, observers drive 20-mile survey routes following unimproved roads, listening for the vocalizations of male prairie grouse displaying on leks (prairie-chickens) or booming grounds (sharptails). The observers stop at 1-mile (~1.6 km) intervals, listen for mating calls during a 5-min period, and mark auditory detections on a map [40]. Surveys begin 1 h before sunrise and finishing no later than one-half hour after sunrise.

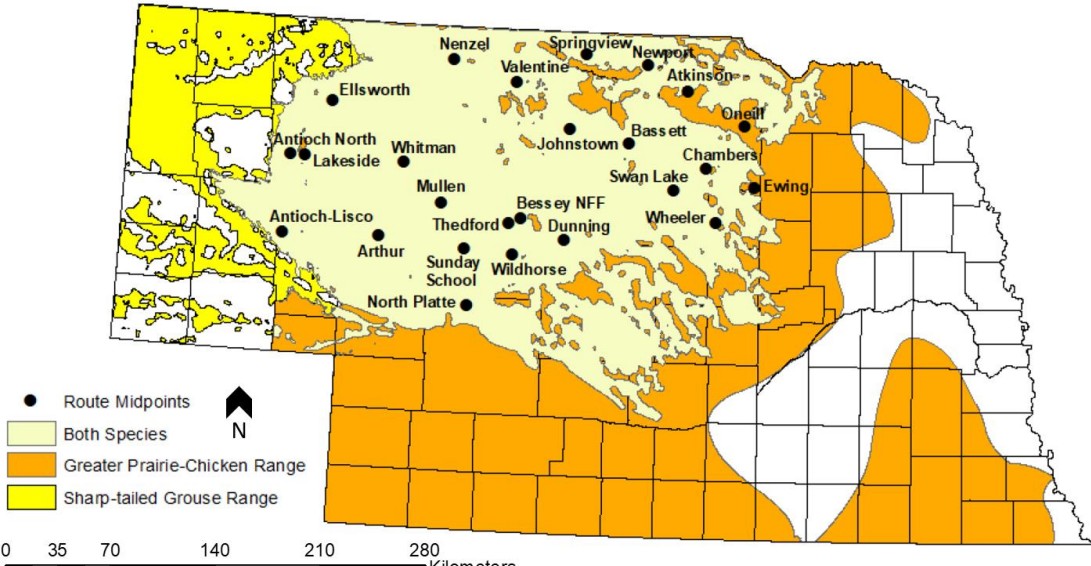

**Figure 1.** Geographic midpoints of the 25 historical prairie grouse breeding ground survey routes that fall within the Sandhills of Nebraska, USA. The boundaries of the Sandhills are roughly approximated by the spatial extent of the beige polygon representing shared prairie grouse range. Adapted from a figure created by J. Dallmann, Nebraska Game and Parks Commission.

Between April 10 and 20, observers visit active breeding grounds identified during initial listening stops, as well as any sites with documented prairie grouse presence in the previous two years. Observers carefully approach the breeding grounds to prevent disturbing the birds, count the total number in attendance, and flush the grouse to confirm the count. Because the number of males on a breeding ground is relatively constant following the appearance of females but female attendance diminishes as hens begin

nesting [41], total lek counts are multiplied by a species-specific coefficient representing the proportion of males likely to be present on the lek on a given Julian date. The total number of males is aggregated for each route to provide an index of species-specific male prairie grouse abundance [30]. Routes are surveyed once each spring, so route-specific count values represent a single sampling occasion.

*2.3. Analyses*

2.3.1. Mean and Transect-Specific Breeding Ground Count Trends

We created a Sandhills-wide, species-specific mean index of abundance by taking the total number of males of a single species observed in a given year and dividing by the number of routes surveyed to control for variation in sampling effort between years. Prairie-chickens have never been observed at 5 locations in the northwestern Sandhills (Figure 1; Ellsworth, Antioch North, Lakeside, Antioch-Lisco, and Whitman), so these routes were excluded from the pooled prairie-chicken time series. To determine if prairie grouse experience density-dependent population growth, we regressed species-specific mean counts at timestep $t + 1$ against mean counts at timestep $t$. Departure from a 1:1 relationship indicates that negative density dependence constrains population growth. In addition to the pooled indices of abundance, we evaluated route-specific count trends for each species. We performed all non-spatial analyses in R (R Version 3.6.2, www.r-project.org, accessed on 24 August 2019).

2.3.2. Growth Rate Trends

We used annual mean counts ($N_t$) to calculate the species-specific population growth rate between years using the formula $r_t = \ln(N_{t+1}/N_t)$, where $r_t$ is the intrinsic rate of population increase from year $t$ to year $t + 1$. We chose to use the intrinsic ($r$) rather than finite ($\lambda$) rate of increase for discrete time series data because it was needed for companion analyses [22]. Using the Tidypop package (Tyre 2019, accessed on 24 August 2019), we constructed species-specific, stochastic exponential growth models with 63 timesteps, reflecting the length of the count timeseries. At each timestep, we randomly sampled a value of $r$ from a normal distribution characterized by the arithmetic mean and standard deviation of $r$ for each species across all years of data. We used the mean count from 1956 as the initial value of $N_t$. Counts in subsequent years are then a function of the count the prior year and a randomly selected value of $r$. We ran each model 100 times and compared counts between the initial and final timesteps to determine if the population had grown or declined. We used a stochastic growth rate model rather than simply characterizing the mean growth rate of each population because the sequence of growth rates can have consequences for the final population size that the mean does not capture [42].

2.3.3. Spatially Explicit Count Trends

We represented each route using its geographic midpoint in spatially explicit analyses. We assigned total counts to the route midpoint because we could not attribute birds to specific breeding grounds or listening stops for a large portion of the historical data. We used inverse distance weighting (IDW) in ArcGIS (Environmental Systems Research Institute, Inc., Redlands, CA, USA) to interpolate prairie grouse counts between route midpoints within a species' range. We applied IDW on a continuous scale, with the upper and lower bounds defined by the upper and lower counts for each species, respectively. We included prairie-chicken survey routes from southeastern Nebraska in our spatially explicit analyses to better capture potential range shifts. However, all other analyses were constrained to routes within the Sandhills where prairie grouse are sympatric.

## 3. Results

*3.1. Breeding Ground Count Trends*

Our analyses included male prairie grouse count data collected between 1956 and 2018 from 25 historical SBGC routes that fell partially or completely within the boundaries

of the Sandhills (Figure 1). All routes had some missing years of data because of changes in survey protocol or NGPC staffing (Figure 2).

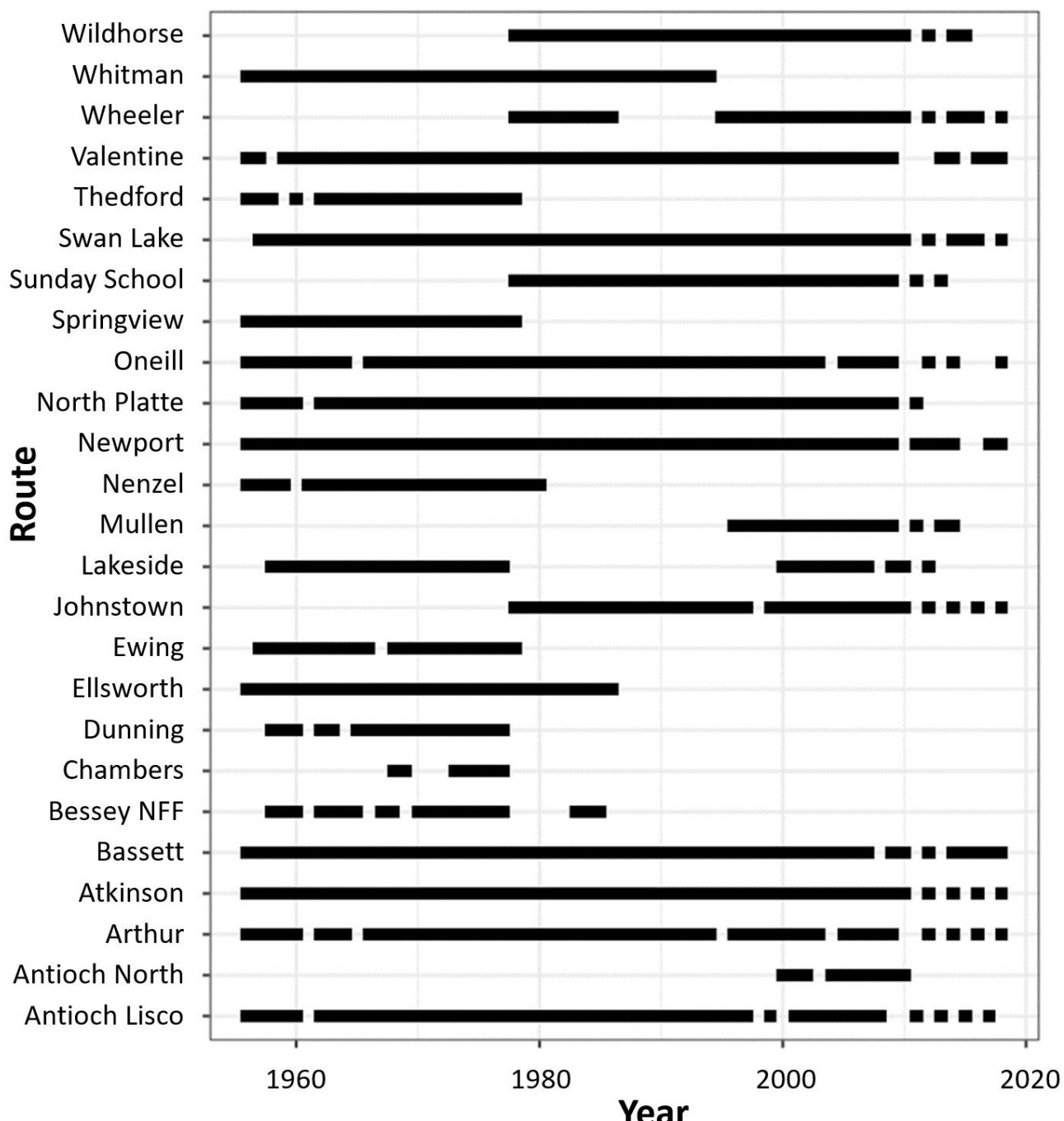

**Figure 2.** Available years of spring breeding ground count data from 1956 to 2018 for each of the 25 historical transects located in the Sandhills of Nebraska, USA. The black rectangles represent the collection of count data for a route in a given year. White spaces indicate an absence of data.

Mean annual count trends reveal that the Sandhills' sharptail population has remained relatively stable since 1956, although it appears that the species may have experienced a slow decline, starting in 1980 (Figure 3). Interestingly, while prairie-chicken and sharptail populations followed similar mean count trajectories from 1956 to 1980, prairie-chicken populations increased dramatically in the Sandhills around the same time that sharptails began to decline (Figure 3).

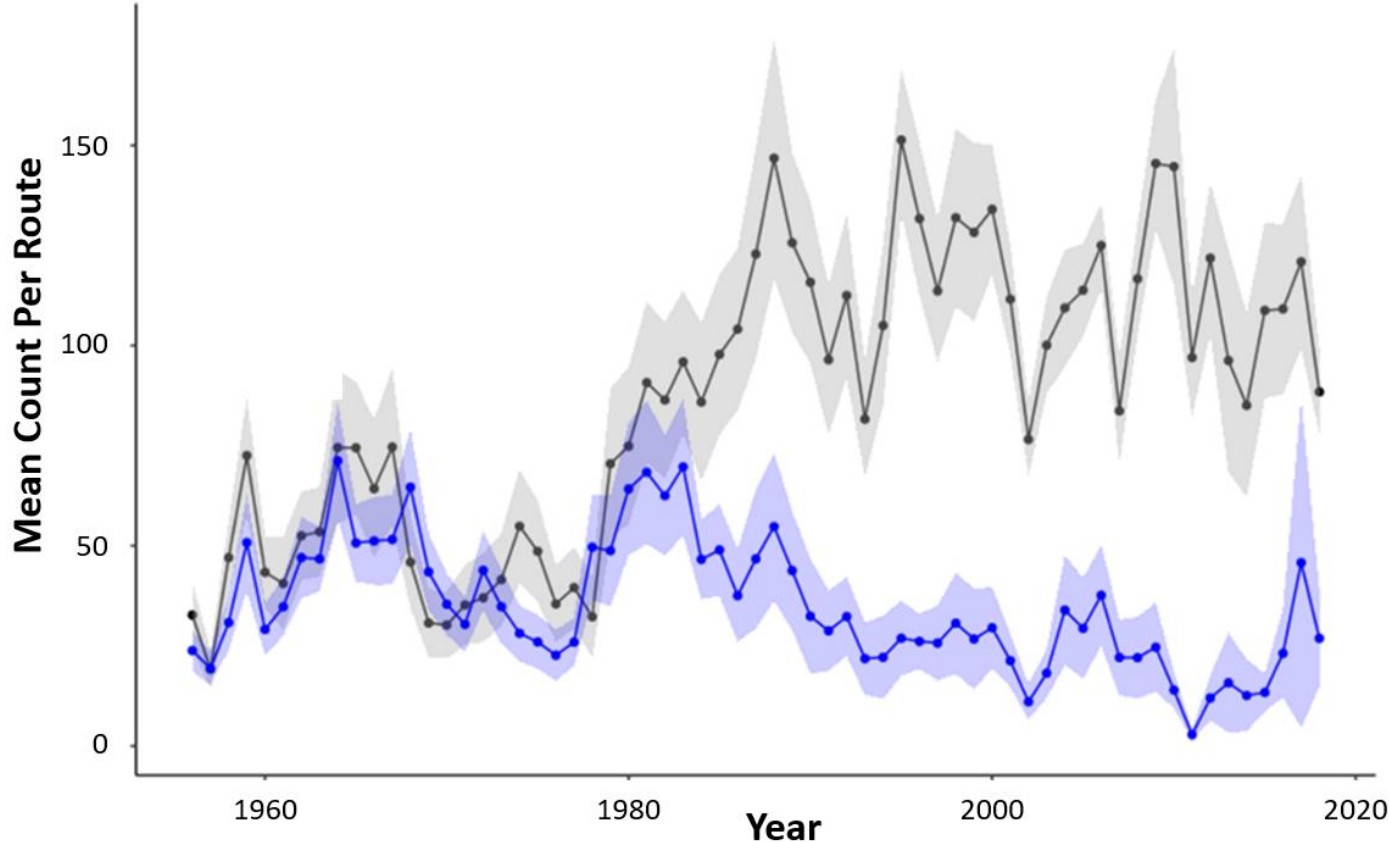

**Figure 3.** Mean number of male sharptails (blue line) and prairie-chickens (grey line) per breeding ground survey route from 1956 to 2018 in the Sandhills of Nebraska, USA. Trend lines are shown with 95% confidence intervals.

Prairie grouse populations were subject to density-dependent growth constraints, and benefitted from positive density dependence when the number of males per route was small, but were subject to negative density dependence as the average number of males increased (Figure 4A,B). Sharptails experienced statistically significant (the 95% confidence interval does not overlap the 1:1 line) positive density dependence when the mean number of males per route was between 0 and 20, and negative density dependence when males per route exceeded 50 (Figure 4A). While prairie-chickens were also subject to density-dependent population growth, they benefitted from statistically significant positive density dependence at count sizes (up to 60 males per route) where sharptails experienced negative density dependence (Figure 4B). Prairie-chickens also were not subject to negative density dependence until mean counts approached 160 males per route (Figure 4B).

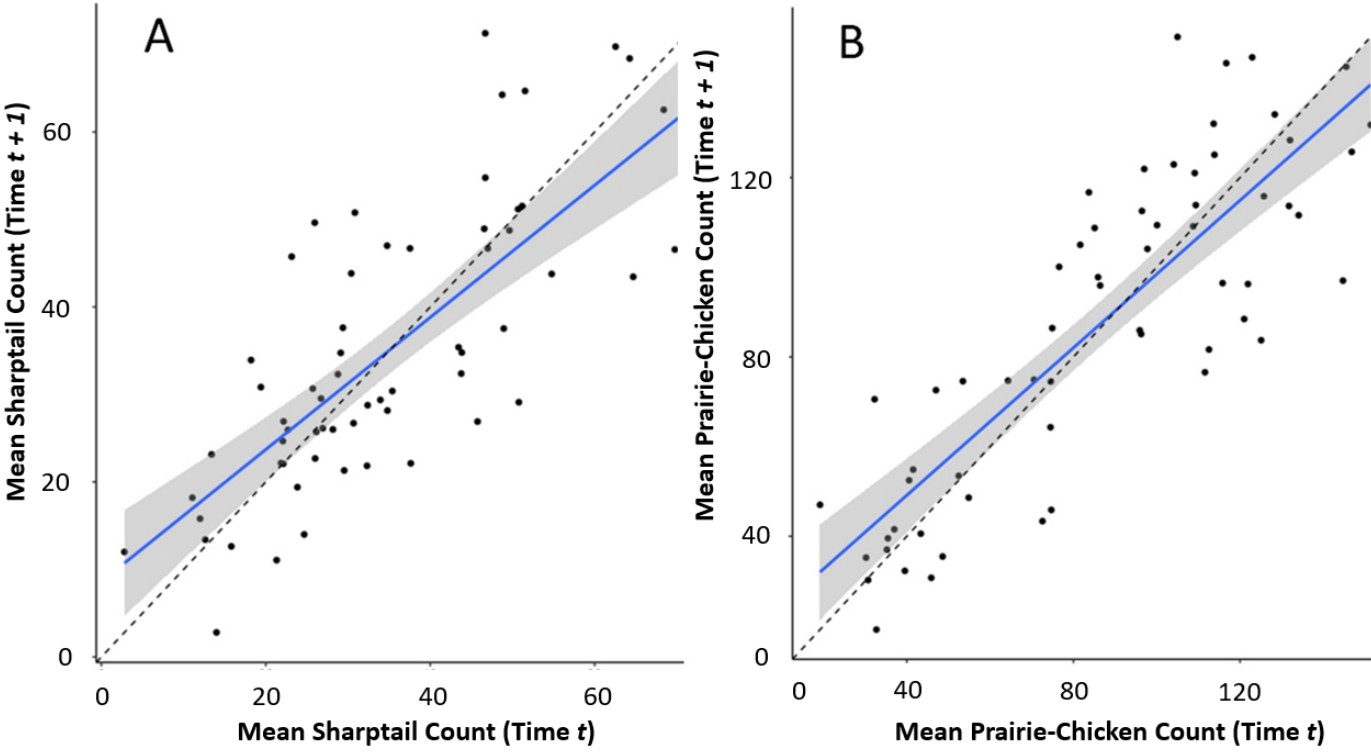

**Figure 4.** Scatter plots of (**A**) sharptail and (**B**) prairie-chicken mean breeding ground counts from consecutive years between 1956 and 2018 in the Sandhills of Nebraska, USA. The dashed lines represent a 1:1 relationship (counts from consecutive years have identical values). Trend lines are linear models with 95% confidence intervals.

Counts varied among survey routes in the Sandhills, with each route exhibiting unique species-specific population trends over time (Appendix A). Although the route-specific analysis was spatially implicit (we did not consider trends in the context of geographic space), to better understand how regional patterns of grouse abundance changed over time, we assigned a subset of 6 transects with nearly complete time series of data to route midpoints (Figure 5). We used the population trends with spatial context to generalize shifts in prairie grouse abundance in the Sandhills since 1956. Sharptail counts have declined in the southern (Figure 5; North Platte) and eastern Sandhills (Figure 5; Swan Lake and Atkinson) since 1956 and in the southwest since the 1980s (Figure 5; Antioch). However, sharptail counts in northcentral Nebraska (Figure 5; Valentine) and prairie-chicken populations in the eastern Sandhills (Figure 5; Swan Lake and Atkinson) have increased since 1956.Prairie-chickens were mostly absent from the northwestern Sandhills until the 1980s (Figure 5; Valentine) and from the southwest until the 1990s (Figure 5; Arthur), but populations have increased in both regions following establishment.

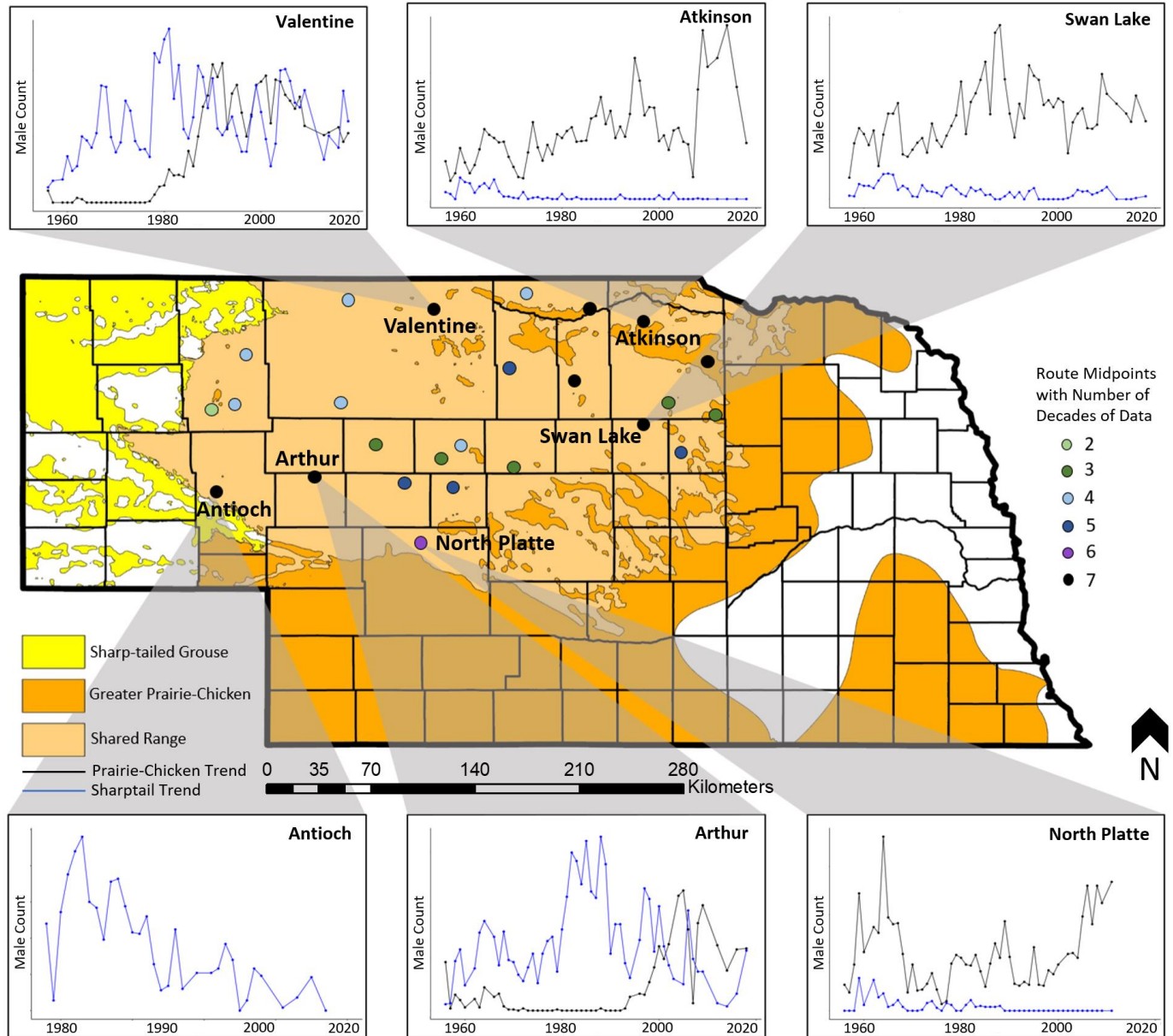

**Figure 5.** A subset of prairie grouse spring breeding ground count time series from 1956 to 2018, by route, in the Sandhills of Nebraska, USA, emphasizing spatial variation in counts over time across the species' respective ranges. Each time series inset depicts the number of males observed on a route, with sharptails represented in blue and prairie-chickens shown in black. The absence of a species from a time series indicates the species was never observed at that location. The colors of the route midpoints represent the number of decades of data collected on each route.

### 3.2. Growth Rate Trends

In 55 out of 100 runs of the sharptail stochastic growth rate model, the average number of males per route was smaller in the final timestep of the simulation than in 1956, suggesting that the sharptail population in the Sandhills was stable or has slightly declined since monitoring began (Figure 6A). The average number of prairie-chickens per route in the final timestep of the simulation was greater than the 1956 counts for all runs of the stochastic model (Figure 6B). Our growth rate simulation corroborates the pooled count data trends with deviating outcomes for the two species of prairie grouse.

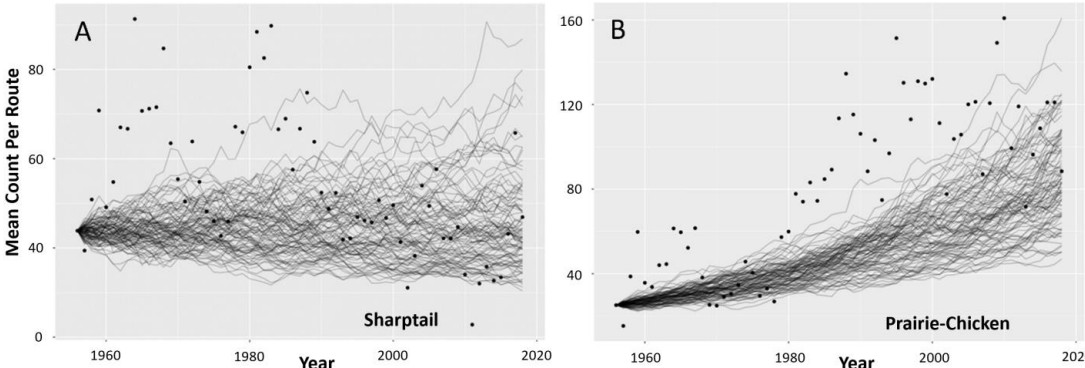

**Figure 6.** Predictions from 100 stochastic runs of (**A**) sharptail and (**B**) prairie-chicken exponential population growth models for the Sandhills of Nebraska, USA. Lines represent the average male count per route over 63 annual timesteps starting from the observed mean count in 1956. Points denote observed annual mean counts from 1956 to 2018.

### 3.3. Spatially Explicit Population Trends

The core of Nebraska's sharptail population, represented by the highest count values, has shifted north and west over time (Figure 7). The movement of the population's center is indicative of a larger range shift as sharptails have retreated from the eastern and southcentral margins of the Sandhills and become more abundant in the north and west (Figure 7). Currently, prairie-chickens have population strongholds along the southern and eastern margins of the Sandhills (Figure 8). Although prairie-chickens were extirpated from southeastern Nebraska around 1900, populations made a resurgence in the 1990s before declining again after 2010 (Figure 8). The population center of prairie-chickens in Nebraska has shifted west over time, but the species has become more abundant throughout the Sandhills (Figure 8).

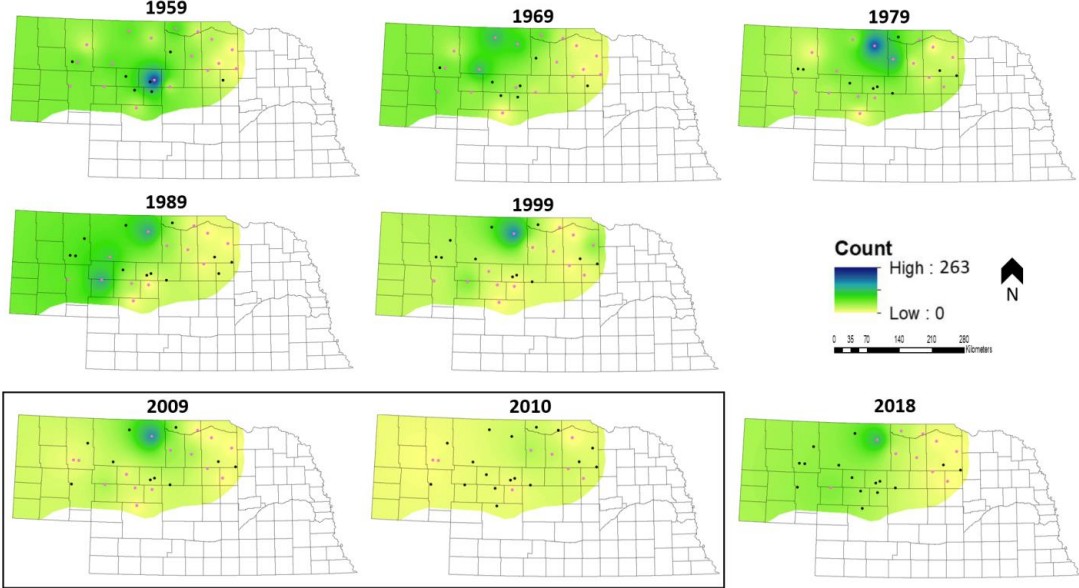

**Figure 7.** Sharptail inverse distance weighted spatial population trends at decadal intervals from 1959 to 2018 in the Sandhills of Nebraska, USA. Pink route midpoints indicate that data were collected at that location in a given year. The two adjacent years encompassed within the black box were included to illustrate the effect of changing the subset of monitored routes between years when little change in population distribution has likely occurred. The time series ends in 2018 with the last year of available data.

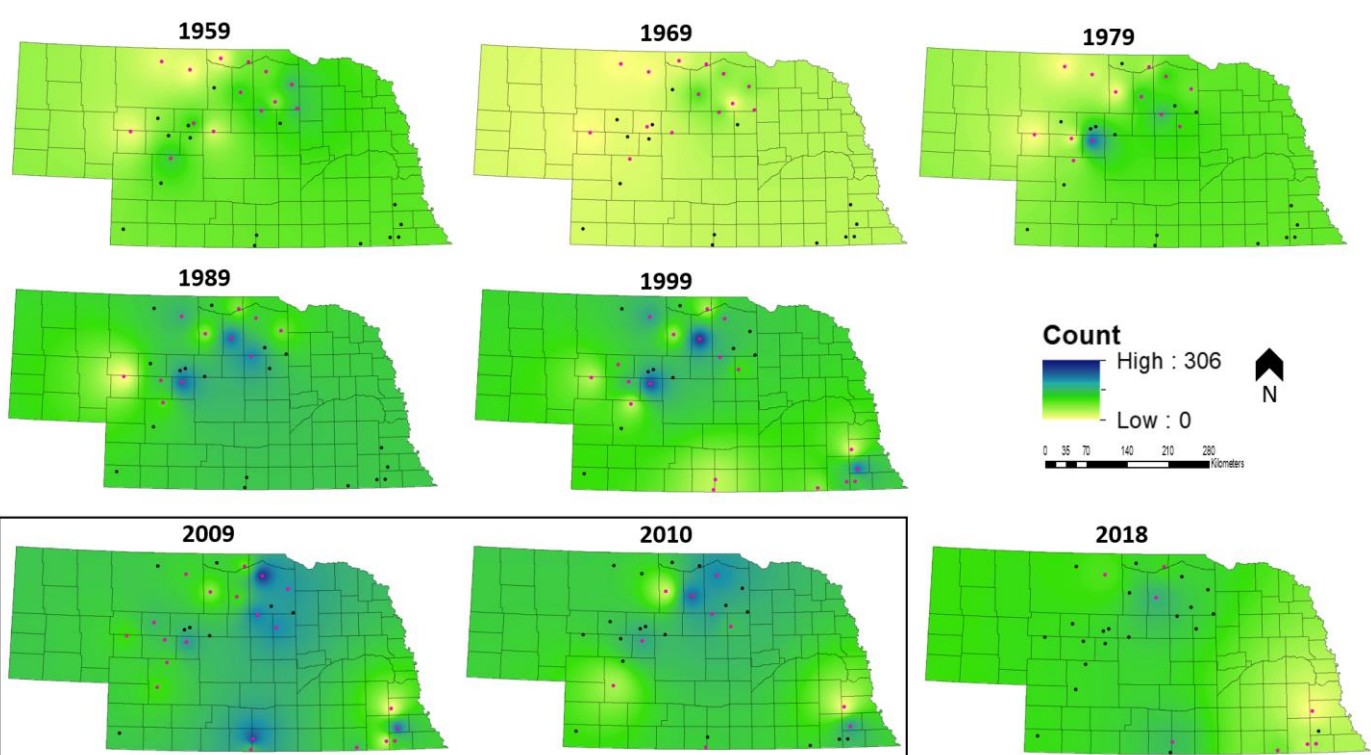

**Figure 8.** Prairie-chicken inverse distance weighted spatial population trends at decadal intervals from 1959 to 2018 in the Sandhills of Nebraska, USA. Pink route midpoints indicate that data were collected at that location in a given year. The two adjacent years encompassed within the black box were included to illustrate the effect of changing the subset of monitored routes between years when little change in population distribution has likely occurred. The time series ends in 2018 with the last year of available data.

## 4. Discussion

### 4.1. Breeding Ground Count Trends

Prairie-chicken populations have increased since 1956, while sharptail populations have remained stable or slightly declined. However, the considerable spatiotemporal variation among survey routes and species' range shifts suggests pooled, non-spatial metrics may oversimplify prairie grouse population trends, potentially delaying critical conservation interventions. Diverging abundance trends, the retreat of sharptails and westward progression of prairie-chickens into the Sandhills, and different thresholds of negative density dependence are all patterns that point to ecological processes inhibiting the effective conservation of these species in shared habitat. However, our results should be interpreted cautiously as we discuss the influences of the sampling protocol and data aggregation on the inferences we have drawn from repurposed data.

Both spatial and aspatial population trends provide evidence that prairie-chickens and sharptails do not behave as a single species in the Sandhills. Prairie-chickens and sharptails do not experience high counts simultaneously on any one SBGC transect, and the species' population centers are spatially disjunct (Figures 7 and 8). Although prior observational studies have found no evidence of interspecific competition between prairie-chickens and sharptails in the Sandhills [43], the last research was conducted 80 years ago, when prairie grouse abundance was significantly lower than observed during our time series [44]. It is possible that the Sandhills provided sufficient resources to support both species at low abundance, but that prairie-chickens and sharptails compete for shared resources at higher abundances. Because competition plays out in a dynamic environment [45], it is also possible that changes in the Sandhills' resources over time came to favor the competitive dominance of prairie-chickens over sharptails.

Environmental shifts that favor one species over another—including in the context of competitive interactions—are indicative of underlying ecological niche differences [46]. An increase in the number of prairie-chickens was not always accompanied by a decrease in sharptail counts (Figure 5; Valentine), which is an unexpected pattern if prairie grouse species compete interspecifically. Sharptails also experience negative density dependence at a count threshold where prairie-chickens are still subject to positive density dependence. Given that these thresholds are measured in common habitat, prairie-chickens and sharptails may not share an ecological niche, as existing management practices in the Sandhills have assumed. While species-specific density dependent constraints could also arise because of behavioral or physiological differences between species, we do not speculate on the importance of these mechanisms because their influence cannot be evaluated using only historical count data. Returning to habitat-related mechanisms, Hiller et al. (2019) found that the breeding niches of sharptails and prairie-chickens in the Sandhills differed, but habitat use was similar during other times of the year [17]. While sufficient environmental heterogeneity could support both species in shared habitat [17], directional shifts in habitat quality toward either the prairie-chicken or sharptail breeding niche could provide a fitness advantage to one species. However, the patterns of habitat selection in Hiller et al. (2019) play out among individuals at a local scale, while our focal patterns are observed among populations at the regional scale [17]. Local breeding season differences in habitat use could produce emergent patterns of regional species distribution [47], but the possibility of unique, scale-dependent, contemporary population drivers should not be discounted.

While the mechanisms driving species-specific trends of abundance and distribution are beyond the scope of inference of this study, population trends suggest that prairie-chickens are doing better than sharptails in the Sandhills. Given the importance of the Sandhills populations for regional conservation efforts—particularly of prairie-chickens—it is imperative for wildlife practitioners to understand that species-specific population drivers likely constrain the management of prairie-chickens and sharptails in shared range.

*4.2. Evaluating Sources of Bias in Count Trends*

The interpretation of the prairie grouse population trends in Section 4.1 assumes that the observed patterns describe biological processes shaping species abundance and distribution. However, SBGC trends may be biased by the observation protocol and method of data aggregation [48]. Before making conservation recommendations for prairie grouse based on the longitudinal SBGC data, it is best practice to ensure that population trends represent biological processes, rather than artifacts of the sampling protocol or analyses [49].

It is difficult to understand how prairie grouse populations have performed in the Sandhills over the past 60+ years based on route-specific abundance trends because of missing data and trend variability among routes, as evidenced by our spatially implicit model. To draw meaningful inference, it is necessary to summarize route-specific trends across space and time. One approach we took was to create a mean index of abundance, following with historical data aggregation practices [30]. A mean annual count model assumes that variation among routes represents noise rather than species biology [50]. However, our spatially explicit model of the count data revealed that variation among routes was not random but exhibited regional trends. The existence of regional trends suggests that a measure of central tendency does not adequately describe the heterogeneity found within the samples.

By considering a finer-scale analysis of patterns of abundance and distribution concurrently, it is possible to start to disentangle the biology from potential bias. Species' range shifts relative to the distribution of survey routes may have caused a negative bias in the mean trend for sharptails and a positive bias for prairie-chickens. Our analyses show that prairie grouse are not evenly distributed throughout the Sandhills. Thus, when the population center for sharptails shifted northwest, the bulk of the population existed in a region with few survey routes. High counts were captured on relatively few routes compared to when the population center was found in the central Sandhills (Appendix A). We suspect

that the mismatch between the distribution of the population and survey routes likely caused the mean count to decline at a magnitude that did not represent the biological trend of the statewide sharptail population. In contrast, as the range center of prairie-chickens advanced west, the number of transects with low or zero counts declined and more routes had high counts (Appendix A). We suspect that the size of the prairie chicken population did not increase in step with the mean count trend over time.

The mean count trends of both prairie grouse species appeared to vary stochastically over time. Although environmental variation may have contributed, we suspect that changes in the number and configuration of routes sampled had a significant role in the observed patterns of variation in the mean count (e.g., review the adjacent survey years encompassed in black boxes in Figures 7 and 8). How many prairie grouse were counted was dependent on where in space the survey transect was positioned—an example of the Modifiable Areal Unit Problem. However, accounting for space, and considering spatial and non-spatial trends in tandem provided a reasonable degree of confidence that the overall direction of the trends was not wrong. Sharptail abundance in the Sandhills has decreased as the species' range has shifted northwest while prairie-chicken abundance has increased, accompanied by a range expansion into the central Sandhills. The results of the stochastic growth rate simulation further support that the trajectory of the trends is accurate, which is the expected degree of precision of an index of prairie grouse abundance for management purposes. If conservation applications of historical count data require greater accuracy than the general trend direction, the magnitude of bias introduced by the sampling protocol could be further explored using simulation models with a known population size.

*4.3. Management Implications and Future Research Directions*

While mean abundance trends may have been adequate to inform prairie grouse hunting regulations applied uniformly across the Sandhills, conservation also requires an understanding of species' stories in space. Conservation interventions, such as habitat restoration, are implemented at local, rather than regional scales [51]. In addition to being useful for validating abundance trends, evaluating how prairie grouse are distributed throughout the Sandhills can help to determine where active management will best serve conservation goals. For successful proactive conservation in a rapidly changing world, it not only matters where populations are currently found, but also where they will be found in the future [52]. The Sandhills form the southernmost boundary of the sharptail's range [7], and the population's center has shifted north in Nebraska since 1956. Given that the Sandhills are likely marginal habitat for sharptails because they are at the edge of the species' range [53] and changes in global climate may already have precipitated the population's northward shift, it may not make sense to manage for sharptails in the Sandhills (although there is a lack of consensus in the literature about the importance of rear-edge populations in the face of climate change [54–56]). Our conclusion about the questionable future value of Sandhills' habitat for sharptails could not be drawn based on abundance trends alone. Knowing that the sharptail population has shifted north can help managers to improve habitat in locations that will actively benefit the species' future persistence. For effective prairie grouse conservation, analytical methods must move beyond a model of mean abundance to explicitly account for the consequences of space when repurposing SBGC data.

Understanding how a route's position in space has influenced spring breeding ground counts is a critical step in repurposing historical count data to address the conservation question of greatest interest for prairie grouse in the Sandhills: What biological processes have shaped species-specific abundance trends? While spatial trends may have complicated the interpretation of changes in prairie grouse populations over time, it suggests an underlying biological explanation for the observed patterns of distribution and abundance. The Sandhills are characterized by east-to-west precipitation [35], vegetation [36], and topographical gradients [33] that likely drive spatial count trends through species-specific

influences on demography. The dependence of demographic outcomes on heterogeneous habitat attributes would confirm that transect placement is subject to the Modifiable Areal Unit Problem, a sampling issue that, if present, should be mitigated in future surveys by accounting for habitat context when placing transects. However, the slow diffusion of dispersing individuals outward from population centers could also have produced patterns of spatial non-independence arising from the movement abilities of the species, rather than environmental drivers [57], and would require different adjustments to survey protocols. Future research should seek to understand how species interactions with their environment through movement and demographic outcomes have shaped observed patterns of abundance and distribution in the Sandhills [58]. In this manuscript, we have demonstrated the utility of historical data for filling existing knowledge gaps for understudied populations of conservation concern. While our analyses focused on repurposing data to understand how prairie grouse populations in the Sandhills have changed over time, future work must explore the potential of historical data to explain why these changes occurred if the Sandhills is to remain home to both prairie grouse species.

## 5. Conclusions

While it is best scientific practice to define research questions prior to collecting data [59], we have demonstrated that old data can tell a new story—a story that it was not originally gathered to tell. However, it is critical to understand how the initial objective of data collection and corresponding survey protocol may constrain alternative data usage. Careful interrogation of the historical SBGC data through a variety of analytical methods revealed bias in abundance trends introduced by sampling and the Modifiable Areal Unit Problem—an issue brought to light by exploring the abundance data in a previously unconsidered spatial dimension. By leveraging the SBGC data to investigate changes in both abundance and distribution through time, we were able to begin to disentangle biology from bias. Prairie-chickens and sharptails have experienced different population trajectories in the Sandhills, suggesting that the two species may have unique habitat requirements restricting co-management in shared range. Given the limited funding available to address questions of conservation concern for prairie grouse and other species, and the proliferation of statistical techniques [60] and geospatial tools accessible to biologists [61], it may be time to reconsider what can be learned from the data in hand.

**Author Contributions:** Conceptualization, D.J.B., L.A.P. and J.P.C.; methodology, D.J.B., J.J.L., L.A.P. and J.P.C.; formal analysis, D.J.B.; data curation, D.J.B.; writing—original draft preparation, D.J.B.; writing—review and editing, D.J.B., J.J.L., J.P.C. and L.A.P.; project administration, L.A.P. and J.P.C. All authors have read and agreed to the published version of the manuscript.

**Funding:** The data used in this manuscript were collected through Nebraska's state monitoring efforts funded by the Federal Aid in Wildlife Restoration Act under project W15R. The data compilation and analyses received no external funding, and D.J.B. was supported by the School of Natural Resources.

**Data Availability Statement:** Data presented in this study are available through the UNL Data Repository at the following URL: https://doi.org/10.32873/unl.dr.20220511.

**Acknowledgments:** Data were gathered by numerous employees of the U.S. Forest Service, U.S. Fish and Wildlife Service, and Nebraska Game and Parks Commission, and were provided to us by the Wildlife Division of the agency. We are grateful to W. Vodehnal, W. Inselman, J. Laux, and J. Dallmann.

**Conflicts of Interest:** The authors declare no conflict of interest.

# Appendix A

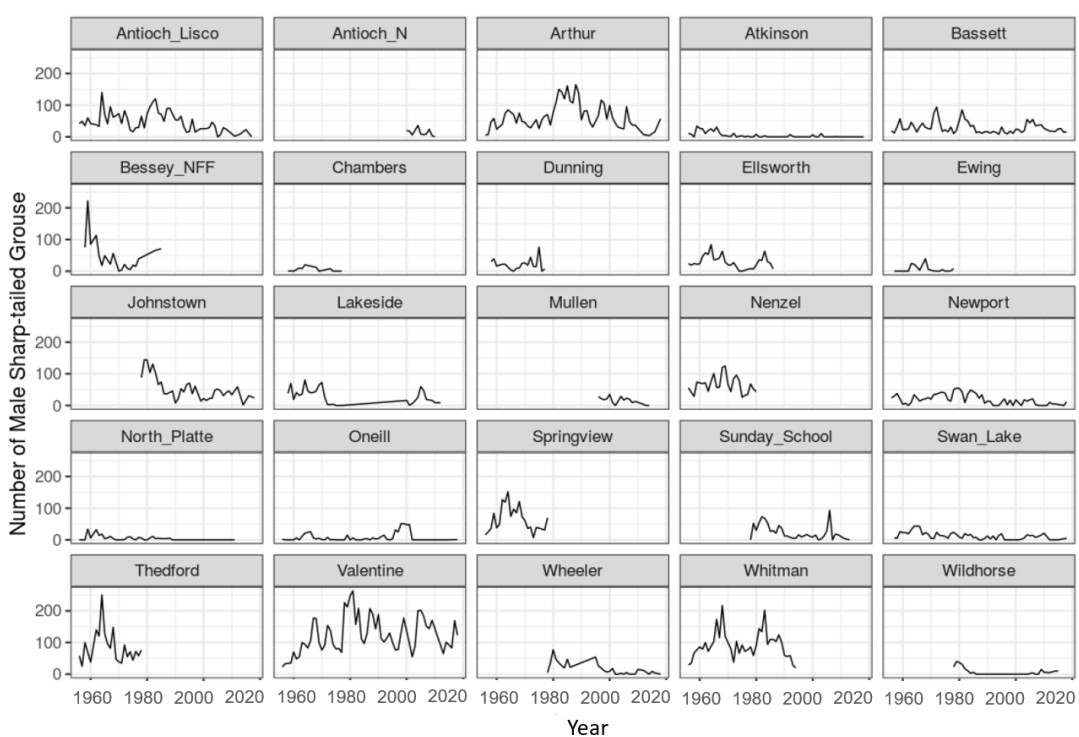

**Figure A1.** Sharptail spring breeding ground counts by 20-mile route from 1956 to 2018 for 25 historical survey transects in the Sandhills of Nebraska, USA.

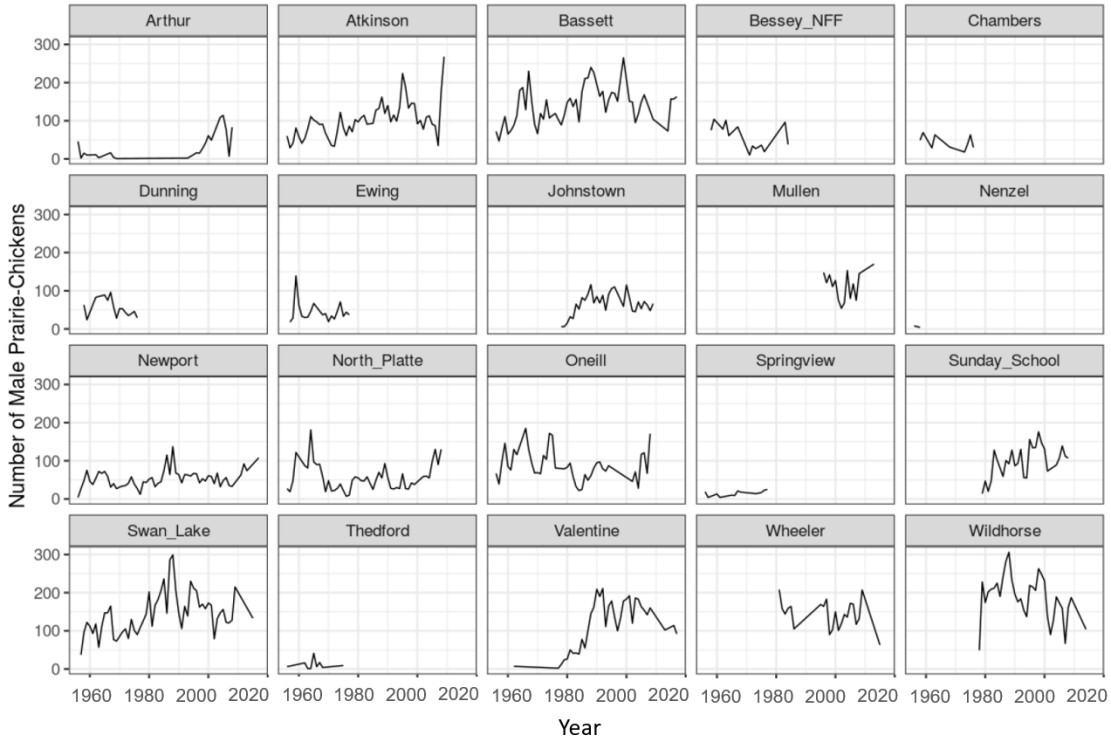

**Figure A2.** Prairie-chicken spring breeding ground counts by 20-mile route from 1956 to 2018 for 20 historical survey transects in the Sandhills of Nebraska, USA.

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
