# Peer review of "Exploring Old Data with New Tricks: Long-Term Monitoring Indicates Spatial and Temporal Changes in Populations of Sympatric Prairie Grouse in the Nebraska Sandhills"

_diversity, doi:10.3390/d15010114_

Round 1
Reviewer 1 Report
Review of Diversity – 2121404 “Exploring old data with new tricks: Long-term monitoring indicates spatial and temporal changes in populations of sympatric prairie grouse in the Nebraska Sandhills”
Thank you for the opportunity to review the manuscript. I found the information timely, relevant, and well presented. Using novel approaches to analyze long-term survey data will aid in conservation and management of many species.
I did not have any concerns with the concept, objectives, analyses, and main conclusions. Much of the manuscript was well written, with only minor editorial suggestions and content comments highlighted below. My principal concern was the construction and structure of the Discussion. While I do not necessarily disagree with the information, I do recommend that much of the Discussion be recast to reduce redundancy, improve flow, and focus on the objectives. I realize that much of the latter portion of the Discussion is focused on the third objective, but there seems to be a disconnect between the third objective and material presented in the Discussion.
My comments on the Discussion material include:
(1) Most journals do not support repeating Results in Discussion, especially citing tables and figures that were previously presented in Results with essentially the same wording related to the findings. I suggest reducing the redundancy between Results and Discussion, removing citations of previously presented figures when there is a lack of new information.
(2) Referring to next steps (e.g., line 410, material in paragraph starting with line 514) needs to be recast to indicate that there is a need for understanding ecological processes that produced the population trends but not directly referencing that this is next step in your research program. Subtle change, I know, but likely best to make this a general goal rather than your next manuscript. I do suggest deleting the sentence that starts on line 410.
(3) Here and throughout the manuscript, carefully consider using among rather than between when describing variation or differences related to routes. Between would be appropriate for comparing 2 routes, whereas among is for comparing >2 routes.
(4) Be careful with the use of terms we and our; appropriate when referencing what you did directly, but not appropriate when making statements of inference or recommendations for management and conservation.
(5) In lines 429-431, you defined spatial autocorrelation. However, I do not think that the definition is correct. Spatial autocorrelation means "the tendency for areas or sites that are close together to have similar values". Also, you can test for autocorrelation. The given definition and associated Discussion does not explicitly address true spatial autocorrelation, but rather a sampling issue. Indeed, I suggest deleting the material on lines 434-472 as it is confusing and the paragraph starting on line 473 seems to summarize the issues in a more concise manner without describing a scenario. At the least, please revise Section 4.2 for clarity and restructure as it is rather difficult to follow.
Other minor editorial suggestions and comments:
Line 12 and throughout, when not at the end of a sentence, please insert a comma after USA (in particular figure captions)
Line 14, delete “the”
Line 27, recall that “data” are plural; replace has with have; line 94 replace was with were; line 101 replace has with have;
Line 32, Keywords – should these be in alphabetical order?
Line 48, change to Greater prairie-chicken; insert ; after scientific name, delete ) (, and combine with hereafter … in ( ) – also on line 49
Line 54, change the to these
Line 66, the paper (19) refers to contiguous not intact grassland and the Sandhills is not the largest in the world, but North America
Line 89, delete the
Line 95, delete third the
Line 104 and throughout, do not start a sentence with an acronym
Line 104, change are to were and insert hyphen 20-mile
Line 109, change hunting to harvest
Line 117, delete second is
Line 120, edit changes to be changed
Lines 134, 136, and 141, change is after objective to was
Line 137, delete the
Line 153 and throughout Study Area – change vegetative (refers to a reproductive strategy) to vegetation
Line 161, change to soil
Line 164, need scientific name for bison; delete the
Line 169, should this be eastern redcedar (no caps)
Line 175 and throughout, do not use st or th associated as superscripts with specific days
Line 176 and throughout, should metric equivalents be given at first mention of distances
Line 186, the statement that number of males on a breeding ground is relative constant throughout the mating season is not quite right – I will accept relatively constant after appearance of females, but until then (usually 2-4 weeks after starting displays), number of males will vary considerably.
Figure 1, and others, check journal format to determine if scale and north arrows are needed
Line 243, change are to were
In Results, tense varies throughout, suggest maintaining past tense throughout
Figure 2, and others, captions must stand alone, so define all acronyms
Line 262, change to prairie-chicken
Figure 3, do not think that the internal labeling of Prairie-Chicken and Sharptail is needed given the figure caption
Line 272, change to density-dependent
Figure 4, need to give years in caption to stand alone
Line 289, citing Figures 9 and 10 is confusing, just cite Appendix
Line 290 change didn’t to did not
Line 292, delete have
Figure 5, these do not look like trends but rather time-series of counts
Figure 6, need to give years in caption to stand alone
Line 329, delete The
Line 336, change almost everywhere in to throughout
Figures 7 and 8, define IDW
Lines 342, 350, change was to were
Line 378, change also known as to i.e.,
Line 385, change Since to Because
Line 386, change habitat to resources
Line 390, change is to was
Line 391, add comma before which
Line 406, insert contemporary before population
Line 410, delete sentence that starts with Our
Line 416, delete third the
Line 426, can you be more specific rather than stating noise
Reviewer 2 Report
I think there will be broad interest in this article given the relative importance of both grouse species to the Great Plains. You did an excellent job of pointing out the uniqueness of the sandhills of Nebraska, with its larger size, more stable or even increasing grouse populations (in the case of prairie chickens), and areas with sympatry. Justification for using old data was solid. Indeed, I don’t think you needed to invest so much of the introduction into the justification.
You presented a very impressive sample size (1956-2018!). You appeared to have been very conscientious in considering the limitations of their data.
Primary objectives basic but important; determine population trends for each species for the sandhills, and compare the trends between species.This was certainly very important considering the status of each species.
Third objective has the broadest implications for conservation beyond the grouse and will make this paper of general interest to management and conservation.
Fig 1. Add N. arrow; please provide for all maps.
Figure 5; Could you label the sites for which you provided additional Figs here? I realize they are labelled on a prior Figure but it would make it easier to understand where the trends are occurring. Again, provide a N arrow. You refer to different trends occurring in distinct regions. Perhaps distinguish these regions on the map for easier viewing?
379-384-rephrase for clarity and brevity; e.g. although prior research has not found interspecific competition to be a factor, but no research has been conducted in the past X number of years etc etc…
394-395; could this also be explained by different social behaviors or other species-specific differences?
425; rather THAN species biology
Section 4.2 Portions of this Discussion are diving into new material that could be enhanced and reworked into the results themselves.
For example, potentially conduct an analysis to determine how fewer transects in the area where the main sharp-tail range shifted to led to an undercount of the population? Likewise, a potential overcount of the prairie chickens over time?
